# Green Synthesis of Characterized Silver Nanoparticle Using *Cullen tomentosum* and Assessment of Its Antibacterial Activity

**DOI:** 10.3390/antibiotics12020203

**Published:** 2023-01-18

**Authors:** John Awungnjia Asong, Ebenezer Kwabena Frimpong, Hlabana Alfred Seepe, Lebogang Katata-Seru, Stephen Oluwaseun Amoo, Adeyemi Oladapo Aremu

**Affiliations:** 1Unit for Environmental Sciences and Management, Faculty of Natural and Agricultural Sciences, North-West University, Private Bag X1290, Potchefstroom 2520, South Africa; 2School of Mathematics, Science and Technology Education, Faculty of Education, North-West University, Private Bag X2046, Mmabatho 2790, South Africa; 3Indigenous Knowledge Systems Centre, Faculty of Natural and Agricultural Sciences, North-West University, Private Bag X2046, Mmabatho 2790, South Africa; 4Döhne Agricultural Development Institute, Plant and Crop Production Research, Private Bag X15, Sutterheim 4930, South Africa; 5Agricultural Research Council–Vegetables, Industrial and Medicinal Plants, Private Bag X293, Pretoria 0001, South Africa; 6Department of Chemistry, Faculty of Natural and Agricultural Sciences, North-West University, Mmabatho 2735, South Africa; 7School of Life Sciences, College of Agriculture, Engineering and Science, University of KwaZulu-Natal (Westville Campus), Private Bag X54001, Durban 4000, South Africa

**Keywords:** antimicrobial, coumarins, Fabaceae, medicinal plants, nanotechnology

## Abstract

Plants serve as an important source of medicine and provide suitable candidate compounds to produce eco-friendly therapeutic agents. They also represent a source of bio-reducer and stabilizer for the development of nanoparticles for downstream applications. This study focused on the green synthesis of silver nanoparticle (CTAgNP) using *Cullen tomentosum* (Thunb.) J.W. Grimes acetone extract and the evaluation of the antibacterial activity of the plant extract and biogenic nanoparticles against two Gram-positive bacteria strains, namely *Bacillus cereus* and *Staphylococcus aureus*. In addition, the phytochemical profile of *C*. *tomentosum* was established using liquid chromatography–mass spectrometry (LC-MS). The antibacterial effect of the extract and CTAgNP was moderate based on the minimum inhibitory concentration (MIC) values obtained. The MIC values of 2.6 mg/mL and 3.1 mg/mL were recorded for *C*. *tomentosum* extract against *B*. *cereus* and *S*. *aureus*, respectively. On the other hand, the CTAgNP had MIC values of 1.5 mg/mL and 2.6 mg/mL against *B. cereus* and *S. aureus*, respectively. The nanoparticle exhibited surface charge of −37 ± 7.67 mV and average hydro-dynamic size of 145 nm. X-ray diffraction illustrates that metallic nanoparticles were formed and had a face-centered cubic structure. Microscopic and spectroscopic techniques revealed that the CTAgNP was covered by a protective shell layer constituted of organic compounds originating from the plant extract. The acetone extract of *C*. *tomentosum* could be useful to the bio-pharma industries in the large-scale manufacture of nanoparticle-based medications to fight against microbes that constitute a threat to the survival of humanity.

## 1. Introduction

The management of infectious diseases has become an issue of global concern due to the development of resistance to antibiotics by pathogens and undesirable side effects of synthetic medications [1,2,3]. As a result, there is a growing need in both developed and developing countries, to increase the monitoring of resistance development by pathogens if a healthcare crisis is to be avoided on a global scale [1,2]. This worrisome situation of consistently evolving antibiotic resistant pathogens (ARPs), has resulted in the renewed interest in metal nanoparticles such as gold (Au), copper (Cu), Zinc (Zn), silicon (Si), and silver (Ag) [4,5,6,7], synthesized from biological organisms (plants, fungi, bacteria, algae, and viruses) for downstream applications in medicine to combat ARPs. These metallics, from a macro lens, have been shown to release reactive oxygen species, which enhances their cytotoxic effects coupled with their subtle oxidation, and liberation of metallic ions have placed them in a favorable position as bactericidal agents in addition to their size, potential to penetrate across the cell membrane and catalyze disturbance of intracellular activities including but not limited to cell permeability, protein synthesis and cell metabolism, which causes cell death [8,9,10,11]. With the emergence of nanotechnology whose monomers are nanoparticles [12,13], various methods of synthesis of these monomers have been advocated. The popular methods include the chemical and physical methods. These are often associated with shortcomings such as cost, reduced efficacy, hazards to the ecosystem (chemical route), enormous energy, and dispelling of radiations (physical route) [14,15]. As such, this necessitates the need for a synthetic route (green synthesis), which will not only mitigate the detrimental effects of the chemical and physical methods but will also be biocompatible, eco-friendly, faster, simpler, use materials that can act as both reducing agent and capping agent, and produce particles of larger size with enhanced surface morphology [7,16,17]. Silver nanoparticles have been synthesized using various approaches including the reduction of silver salts (silver nitrate) by sodium borohydride, photochemical reduction, electrochemical reduction, and plant material (green synthesis) amongst others [18,19,20,21]. Silver nanoparticles (AgNPs) are readily available, with relatively high potency against bacteria when applied in bioburden management, and with the possibility of having limited side effects, and the ARPs have reduced possibilities of developing resistance to AgNPs [22,23,24,25]. Furthermore, AgNP antimicrobial strength has been attributed to other AgNP factors including shape, morphology, surface modification, stability and interaction with their environment [9]. Hence, AgNPs from various sources are being favored for downstream health applications. Interestingly, the efficacy of nanoparticles developed from plants such as *Rhodomyrtus tomentosa* Hassk. and *Tagetes erecta* L. using their acetone extracts have been reported against *Escherichia coli, Staphylococcus aureus*, *Bacillus cereus,* and *Pseudomonas aeruginosa* [18,19].

*Cullen tomentosum* J.W. Grimes (family: Fabaceae) is locally known by the Batswana people in South Africa as “Mojakubu” and has three synonyms, namely *Cullen obtusifolia* (DC.) C.H. Stirt., *Psoralea obtusifolia* DC., and *Trigonella tomentosa* Thunb, as described by von Staden [26]. *Cullen tomentosum* has psychotropic properties with an effect on human mind-set. In traditional medicine, its leaves and stem are smoked and used as narcotics while macerations of the whole plant have been used for the treatment of rashes and sores on the skin, and as magic and charm in folkloric medicine [27,28,29]. The present study was aimed at synthesizing green AgNP using acetone extract of *C*. *tomentosum* and evaluating its antibacterial potency in comparison to the chemically profiled acetone extract.

## 2. Results and Discussion

### 2.1. Characterization of the Cullen tomentosum Silver Nanoparticle

#### 2.1.1. UV-Visible Spectrophotometry

The UV-Vis spectrum of a solution containing the biogenic CTAgNP was acquired within the range of wavelength 200–800 nm (Figure 1). This monitored the reduction of Ag+ ions to zerovalent Ag in the presence of phytocompounds identified in *C*. *tomentosum* acetone extract. It is well-argued in the literature that the change in color observed with solution of Ag+ ions in the presence of electron donors is due to the vibrations of the surface plasmon resonance (SPR) in AgNP [30,31]. Therefore, the appearance of a strong peak at 418 nm, which lies between 418–428 nm of the characteristic SPR band for silver, suggests the formation of CTAgNP with enhanced particle size. The absence of peaks between 296 and 632 nm is proof that there was no aggregation of nanoparticles. A similar peak was reported by Rao et al. [32] for the development of AgNP using *Diospyros paniculata* methanol extract.

#### 2.1.2. Dynamic Light Scattering (DLS) 

The DLS analysis of CTAgNP was performed in water and the results presented in Figure 2. This revealed hydrodynamic size of 145 nm and PDI of 0.236. Similar results were obtained by Banerjee et al. [30] from leaf extract of *Mentha arvensis*, which recorded AgNP with a hydrodynamic diameter of 145 nm and PDI value of 0.226. PDI values below 0.3 are more often considered acceptable for drug delivery systems and these infer a homogenous suspension. The zeta potential recorded −37.2 ± 7.67 mV, which indicated a relatively stable CTAgNP. High magnitude of negative charge at the surface of CTAgNP suggests a strong Coulombic repulsion among the nanoparticle formed [33]. This observation is presumably ascribed to the presence of phytochemicals at the surface of the developed silver particle, which contributes to increasing its stability.

#### 2.1.3. FTIR Spectroscopy

As shown in Figure 3, the FTIR spectrum of the CTAgNP displayed characteristic peaks at 618, 1032, 1425, 1652, 2983, and 3332 cm^−1^ assigned to C-C skeletal vibration [34], C-O stretching of secondary alcohol [35], N-O stretching of the nitro groups [36], C=O stretching of amide [37], -CH_3_ asymmetry stretching [38], and OH/NH stretch [39], respectively. These supported the assumption that the AgNP obtained were enclosed by phytocompounds. It is anticipated that the functional groups observed also contributed to the bio-reduction of Ag^+^ ions. These could aid as robust binding sites and electron-rich spots for the conversion of silver ions to AgNP. Findings from the current study agree with existing reports that the bio-reduction Ag ions served as binding sites for the conversion of Ag ions into AgNP [40,41,42].

**Figure 2 antibiotics-12-00203-f002:**
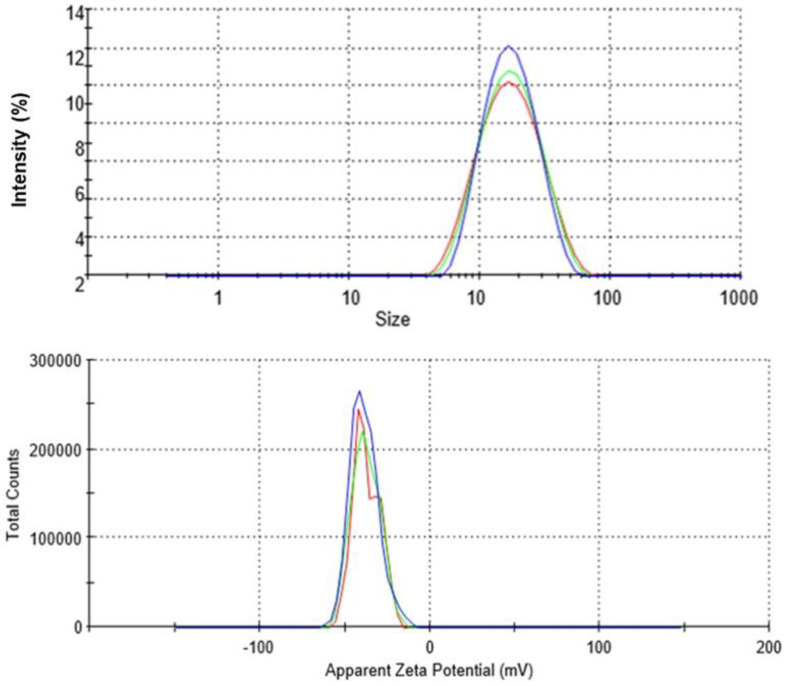
Size distribution and zeta potential of *Cullen tomentosum* silver nanoparticle (CTAgNP).

#### 2.1.4. SEM-EDX and TEM Analyses

A close view of the SEM image of the CTAgNP revealed a smooth surface of aggregates and spherical-like nanoparticle (Figure 4A). Recently, Basavarajappa et al. [40] and Femi-Adepoju et al. [41] observed a comparable morphology for the developed AgNP in *Passiflora vitifolia* and *Gleichenia pectinata* extracts, respectively. EDX was employed to examine the elemental constituents of the CTAgNP and their relative abundance. This investigation revealed intense peak of Ag (17.2%), along with C (62.74%), O (18.95%), and Cl (1.12%) (Figure 4B). The appearance peaks corresponding to the presence of C, O, and Cl corroborate the development of organic-coated AgNP [43]. The topological feature of CTAgNP was analyzed using TEM and this confirmed that the developed nanoparticle was spherical, with an average size of 12.6 nm (Figure 4C). It was evident that this size differs when compared to the size determined using DLS. Similar observation of a relatively higher particle size estimation by DLS when compared to that determined by TEM was reported by Al Hagbani et al. [44]. This may be explained on the basis that DLS determines the size of nanoparticles in a hydrated state whereby a solvent layer shields the surfaces of nanoparticles. Hence, the size constitutes both the inner core and the sheath. On the other hand, TEM analysis is based on a dry state of the nanoparticle and the estimated size does not include the solvent sheath [45,46]. A related study also described the spherical shape of the biosynthesized silver particles with an average size of 7–27 nm [47]. Likewise, spherical AgNP of 10 nm were reported to form in aqueous sorghum bran extracts [48].

#### 2.1.5. X-ray Diffraction (XRD)

The phase, alignment, and particle size of the CTAgNP were assessed using XRD analytical techniques. This analysis provided an understanding of the specific formation of crystalline nanoparticles by the appearance of sharp peaks in the diffractogram [48]. A set of major peaks at 2θ 38.5°, 46.5°, 67.5°, and 77.0°, consistent with the Miller indices (111), (200), (220), and (311), respectively, were detected (Figure 5). These data matched with the standard database of the Joint Committee on Powder Diffraction Standards (JCPDS, no 4.0783). This confirmed the face-centered cubic (FCC) crystalline Ag. The results obtained in this study is in line with previous studies of FCC AgNPs formed in medicinal plants [49,50]. The unassigned peaks observed are certainly due to the bioorganic phases that converged on the surface of CTAgNP [51]. The average size of the CTAgNP using Scherer’s equation [D = (0.89λ/β COS θ), where “D” represents the mean of the AgNPs, “λ” signifies the wavelength of value (0.15418 nm), “β” denotes full width at half maximum (FWHM) peak and θ indicates Bragg diffraction angle [52], was 13 nm. Interestingly, this size was in good agreement with the TEM result.

### 2.2. Antibacterial Potency of Extracts and Cullen tomentosum Silver Nanoparticle (CTAgNP)

As shown in Table 1, the acetone extract had moderate potency against *B*. *cereus* and *S. aureus* using the classification by Katerere and Eloff [49]. The authors classified the potency of plant extracts based on the MIC values as noteworthy (≤1 mg/mL), moderate (≥1–8 mg/mL) and weak (≥8–12.5 mg/mL). The MIC values of the extract against *B. cereus* and *S. aureus* were 2.6 and 3.1 mg/mL, respectively. The result of this study is not isolated, as previous studies have recorded moderate antimicrobial activity of some members of Fabaceae (such as *Erythrina caffra* Thunb) to *S*. *aureus* and *B*. *cereus* [50]. Obistioiu et al. [51] indicated that some members of the Fabaceae such as *Coronilla varia, Melilotus officinalis, Ononis spinosa,* and *Robinia pseudoacacia* had limited inhibitory effect against *S*. *aureus*. The CTAgNP was more potent than the acetone extract with an MIC of 1.5 mg/mL for *B. cereus* while for *S*. *aureus*, the MIC value was 2.6 mg/mL. The improved potency of CTAgNP compared to the extract was noteworthy given that it possesses a hefty ratio of surface area to volume ratio, ability to deform cell membrane and increase permeability as well as a high aptitude to penetrate the bacterial cell wall. This may be due to the shape and size of the nanoparticle, which enhance its ability to interfere with intracellular processes including protein synthesis [9,52] when compared to the plant extract.

We herein for the first time reported the antibacterial activity of both the extract and CTAgNP from *C. tomentosum* against the two tested pathogens. Though moderate, the current findings offer basis for further studies using other microbes. This is particularly based on the existing diverse local use of the plant in traditional medicine. It can be further explored as a candidate in the development of moderate therapeutic agents for individuals who are very sensitive or allergic to drugs with high spectrum potency.

### 2.3. Phytochemical Profiling

The LC-MS analysis of the extracts revealed twenty-four major peaks corresponding to compounds in the whole plant extract (Appendix B, Table 2 and Appendix A). The compounds were further identified by comparing their mass spectral, peak area, peak area (PA) percentage and peak retention time (RT) to that of the m/z cloud library. The least peak area percentage (0.252%) recorded, corresponded to the compound 5,7-dihydroxy-6-methoxy-2-(4-methoxyphenyl)-4H-chromen-4-one with the molecular formular (MF): C_17_H_14_O_6_ and molecular weight (MW): 314.29 g/mol using PubChem database [53] while the highest peak area (11.988%) had corresponding best match to the compound 1-(2-((4-(2-methoxyphenyl)piperazin-1-yl)methyl)-4-methylthiazol-5-yl)ethenone. However, the PubChem database search could not reveal any information related to the molecular formula and molecular weight, indicating that it is probably a novel phytocompound that has not been previously identified in any plant, along with five other compounds with peak area percentages: 1.654%, 2.77%, 1.414%, 4.224%, and 0.907% (Table 2). These compounds could not be identified using the best match criteria on m/z cloud database though their structures had best match equivalent. Other database tools such as PubChem were unsuccessful. 

The phytochemical analysis revealed the presence of cyanidin in *C*. *tomentosum* whole plant extract. This is a water-soluble anthocyanin that is associated with anticancer and antioxidant properties [54]. This finding reveals *C*. *tomentosum* as a new source of cyanidins. A closer look at the peak area percentages reveals that more than 50% of the compounds in the plant fall within five functional groups (Table 2; No 1, 2, 22, 23, and 24) whose peak area percentages sum up to more than 50%. Another compound, 4-((3,3-dimethyloxiran-2-yl)methoxy)-7H-furo[3,2-g] chromen-7-one, belongs to the coumarins known within the subclass Furanocoumarin (Table 2, No 19) which are synthesized through the phenylpropanoid pathway. They have a wide spectrum of biological activity, ranging from antimicrobial, anticoagulant, to anticarcinogenic properties; hence, they have a high potential to fight infections in living tissues [55,56]. Based on existing evidence, the observed antibacterial effect against the tested pathogens may probably be because of the phytochemicals, synergistic activity of a couple of them and the silver containing compounds in the plant. This clearly provides an indication of the need for further investigation on *C*. *tomentosum* for the presence of therapeutic agents. 

## 3. Materials and Methods

### 3.1. Collection of Plant Material and Extraction

The whole plant of *Cullen tomentosum* was harvested in Mafikeng (latitude 25° 51′ 0′′ S and longitude 25° 37′ 60′′ E), South Africa. Voucher specimen (code ja021) was prepared and deposited at the South African National Biodiversity Institute (SANBI) in Pretoria. Freshly harvested plant material was oven-dried at 40 °C to constant dryness for 72 h, powdered using a blender and stored in a sealed container at room temperature. The powdered plant material was extracted in acetone (1:20 *w*/*v*) at room temperature for 1 h with continuous stirring on a magnetic stirrer and later sonicated for 1 h in ice-cold water.

The extract was filtered using Whatman No. 40 filter paper (Whatman^®^ Schleicher & Schuell, London, UK) and concentrated under vacuum using a rotary evaporator at 40 °C. The concentrate was kept in a pre-weighed glass vial and dried at room temperature under a stream of air to a constant weight and stored at 10 °C in the dark till downstream application. All reagents used were of analytical grade.

### 3.2. Synthesis of Cullen tomentosum Silver Nanoparticle (CTAgNP)

The CTAgNP was synthesized from the whole plant acetone extract of *C. tomentosum* following the protocol described by Pethakamsetty et al. [57], as applied in the green synthesis of AgNP from *Diospyros sylvatica* root with slight modifications. Briefly, silver nitrate (AgNO_3_, 0.01 mol/L) and acetone extract of *C. tomentosum* (50 mg/mL) at a ratio of 9:1 (*v*/*v*) were mixed in a beaker and incubated at room temperature overnight with continuous stirring. Following incubation, the formation of nanoparticle was marked by the appearance of a characteristic reddish-brown coloration. This was then parted by centrifuging at 5500× *g* rpm for 5 min. The concentrate was washed thrice with deionized water and dried in a muffle furnace for approximately 18 h, at room temperature. The solid material was labelled as CTAgNP and kept at room temperature in Eppendorf tubes for further use.

### 3.3. Characterization of the Cullen tomentosum Silver Nanoparticle (CTAgNP)

The developed nanoparticle (1 g/mL in ethanol) was characterized using different techniques, including, ultraviolet (UV)-visible spectroscopy (PerkinElmer spectrometer, Lambda 365) for optical absorption and DLS (Malvern instrument, Nano ZS 90) for hydrodynamic size, zeta potential and polydispersity index (PDI) measurements. The morphological characteristics were assessed using scanning electron microscopy (SEM) and transmission electron microscopy (TEM) imaging. The crystallinity, surface functional groups, and elemental composition were discussed based on X-ray diffractogram (Diffractometer, Bruker D8 advance), FTIR (Spectrophotometer, Bruker Alpha), and energy dispersive X-ray (EDX) spectral interpretation, respectively.

### 3.4. Antibacterial Assay

*Bacillus cereus* (ATCC^®^10876™) and *Staphylococcus aureus* (ATCC^®^11632™) were used to determine the minimum inhibitory concentration (MIC) of the acetone extract and synthesized CTAgNP using the microtiter plate dilution technique [58]. *Staphylococcus aureus* is a facultative anaerobe and among the leading pathogens clinically associated with nosocomial infections. With the current proliferation of antimicrobial resistance, *S*. *aureus*, with its capability to assault different tissues, has emerged as a microbe of global concern in medical facilities particularly around intensive care units [59]. On the other hand, *B*. *cereus* is an endospore-forming aerobic or facultative anaerobe, whose pathogenic activities are clinically associated with food poisoning, gastrointestinal infections, skin infections, cutaneous infections—especially in people diagnosed with underlying conditions including diabetes—and traumatic injuries [60,61,62,63]. The aforementioned factors guided the choice of these microbes for the current study.

Both microbial cultures were revived by streaking on Mueller Hinton (MH) agar plates and incubated at 37 °C for 24 h. A positive colony of each bacterium was inoculated in sterilized 10 mL MH broth and nurtured at 37 °C on a shaker for 24 h. The overnight cultures were diluted to a ratio of 1:100 (*v*/*v*). Doxycycline (Sigma-Aldrich, Hamburg, Germany) was used as the positive control. An aliquot of 100 µL of each bacterial strain was added to each well of the microliter plate. A concentration of 50 mg/mL for both extract and CTAgNP in DMSO were put in well A (first well) and two-fold serially diluted with sterile distilled water to well H (last well). Bacteria-free MH broth (blank) and DMSO (which was similarly serially diluted) were used as negative controls. The plates were para-filmed and incubated at 37 °C overnight, after which 50 µL (0.2 mg/mL) of *p*–iodonitrotetrazolium chloride (INT; Sigma-Aldrich, Hamburg, Germany) was added to each well. The microtiter plates were further incubated at 37 °C for an additional 1 h to detect growth. Generally, INT is a colorless compound that is reduced to pink to red coloration by biological activity of microbes [58,64]. Appearance of pink coloration in a well signifies bacterial growth while lack of color change indicates inhibition of bacteria growth. The extract concentration in the last well without any color conversion was recorded as the MIC. The experiment was done in triplicate

### 3.5. Phytochemical Profile Using Liquid Chromatography–Mass Spectroscopy (LC–MS)

Acetone extract of *C. tomentosum* was characterized using LC-MS instrument (LC-MS-2020, Shimadzu Scientific Instruments, Tokyo, Japan). The instrument was equipped with an electrospray ionization (ESI) source operating in negative and positive modes (*m/z* 100–1200), nebulizing gas (1.5 ℓ/min), DL temperature (250 °C), heat block temperature (200 °C) and detector voltage (0.19 Kv). The chromatographic system consisted of a reversed-phase shim-pack C_18_ column (5 µm, 250 mm × 2.1 internal diameter) maintained at a constant temperature (30 °C) using an oven. The mobile phase comprises mixture A (10-mM ammonium formate in 90% acetonitrile: water, *v/v*) and mixture B (0.1% formic acid in acetonitrile, *v/v*). The extract was dissolved in LC-MS-grade acetonitrile and 2.0 µL was injected into the chromatographic system. An isocratic elution was achieved with 30% mixture A and 70% mixture B at a flow rate of 200 µL/min [65]. Authenticated chemical standards (reserpine and nitrophenol) were analyzed at the same chromatographic conditions to calibrate and tune the MS. Water, ammonium formate and acetonitrile used to prepare mobile phase were of LC, UV grade and LC-MS grade (Macron, fine chemicals). Data acquisition and MS spectral analysis were executed using LabSolution software (Shimadzu) and recorded as absolute intensity and *m/z* values. The phytochemicals in the acetone extract of *C. tomentosum* were tentatively identified by exporting MS data (*m/z* values and corresponding absolute intensities) for each peak and loading into *m/z* cloud software to search for possible compounds matching the MS fingerprint in the library (HighChem LLC, 2013–2020). The compounds with the highest best match percentage were recorded.

## 4. Conclusions

The findings of this study revealed the phytochemical constituents of the acetone extract, and antimicrobial potency of both the extract and the biosynthesized CTAgNP. The synthesis of the novel CTAgNP, which was characterized using UV-vis spectrum, DLS, XRD, FTIR, SEM, and TEM analyses, is eco-friendly and cost effective. Both the whole plant acetone extract and the CTAgNP exhibited moderate antimicrobial potency against the studied bacteria pathogens. This suggests the potential of both the whole plant acetone extract and the CTAgNP as antimicrobial agents with applications in different fields including biomedicine. The antibacterial activity of the nanoparticle was more potent in comparison to that of the whole plant acetone extract. However, there is the need to conduct further pharmacological studies to critically understand the mechanism of action including toxicology aspects of the biosynthesized CTAgNP.

## Figures and Tables

**Figure 1 antibiotics-12-00203-f001:**
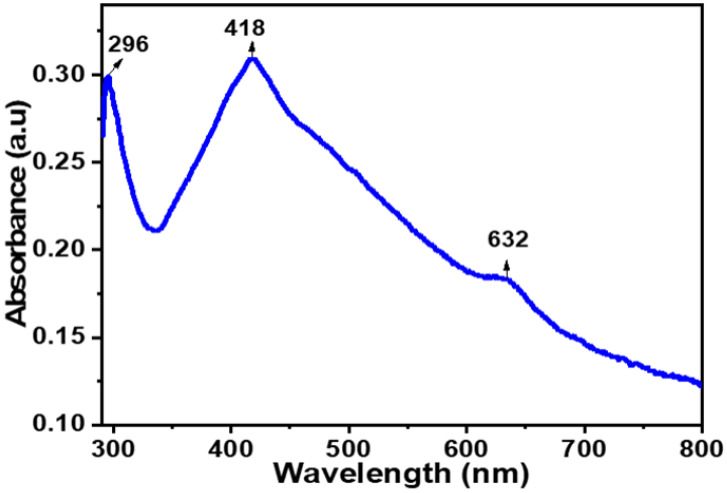
Ultraviolet (UV)-visible (UV-Vis) spectrum of an aqueous solution containing *Cullen tomentosum* silver nanoparticle (CTAgNP).

**Figure 3 antibiotics-12-00203-f003:**
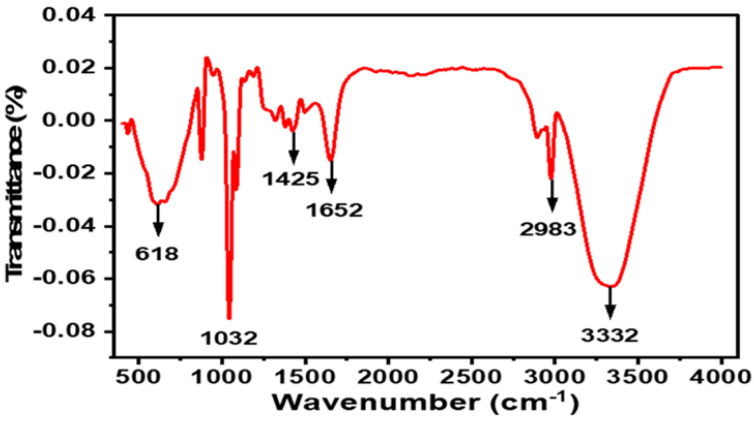
Fourier-transform infrared spectroscopy (FTIR) spectrum of *Cullen tomentosum* silver nanoparticle (CTAgNP).

**Figure 4 antibiotics-12-00203-f004:**
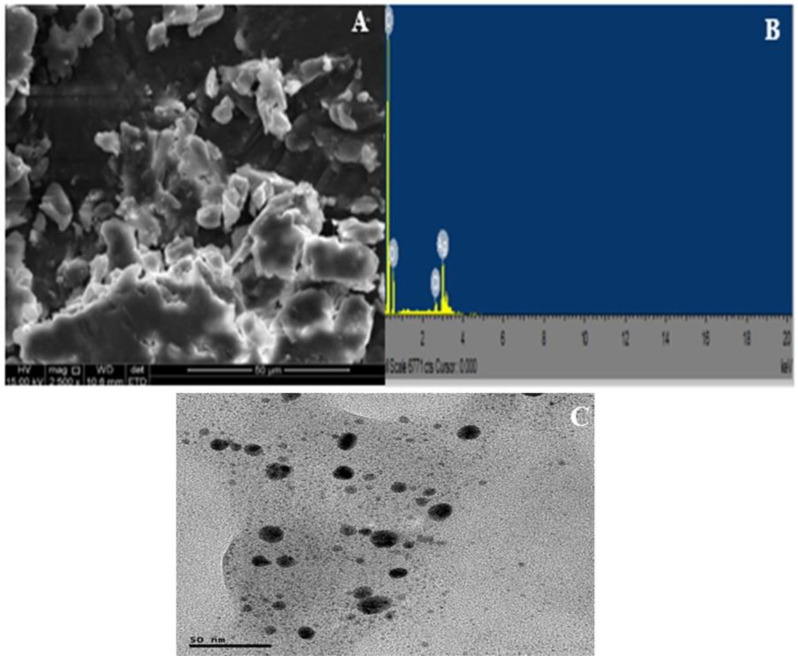
(**A**) Scanning electron microscopy (SEM) image, (**B**) energy dispersive X-ray (EDX) spectrum, and (**C**) transmission electron microscopy (TEM) micrograph of *Cullen tomentosum* silver nanoparticle (CTAgNP).

**Figure 5 antibiotics-12-00203-f005:**
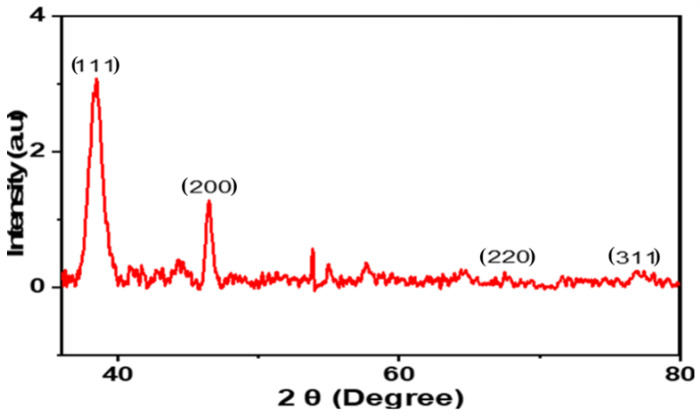
X-ray diffraction (XRD) pattern of *Cullen tomentosum* silver nanoparticle (CTAgNP).

**Table 1 antibiotics-12-00203-t001:** Antibacterial effect of *Cullen tomentosum* acetone extract and resultant silver nanoparticle.

Sample	Minimum Inhibitory Concentration (mg/mL)	
	*Bacillus cereus*	*Staphylococcus aureus*
Acetone extract	2.6	3.1
Silver nanoparticle (CTAgNP)	1.5	2.6

Doxycycline (used as a positive control) had minimum inhibitory concentrations of 0.78 mg/mL and 1.56 mg/mL against *Bacillus cereus* and *Staphylococcus aureus*, respectively.

**Table 2 antibiotics-12-00203-t002:** Phytocompounds identified in the acetone extract of *Cullen tomentosum* following liquid chromatography–mass spectrometry (LC-MS) analysis.

Peak Number	RT *	PA ^#^ (%)	Structure of Compound	Name of Compound (*m*/*z* Cloud Library)	Similarity Index (%)
1	2.326	11.63	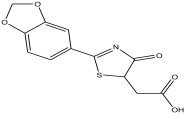	2-(2-(benzo[d][1,3] dioxol-6-yl)-4,5-dihydro-4-oxothiazol-5-yl)acetic acid	81.3
2	2.552	11.988	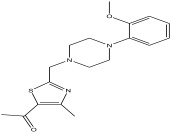	1-(2-((4-(2-methoxyphenyl) piperazin-1-yl)methyl)-4-methylthiazol-5-yl)ethanone	82.2
3	3.59	4.545	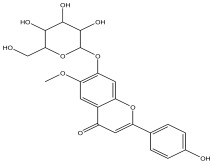	2-(4-hydroxyphenyl)-6-methoxy-7-(tetrahydro-3,4,5-trihydroxy-6-(hydroxymethyl)-2H-pyran-2-yloxy)-4H-chromen-4-one	56.8
4	4.356	1.053	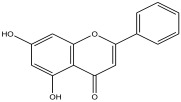	5,7-dihydroxy-2-phenyl-4H-chromen-4-one	90.3
5	5.577	0.588	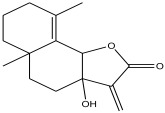	3,3a,4,5,5a,6,7,8-octahydro-3a-hydroxy-5a,9-dimethyl-3-methylenenaphtho[1,2-b]furan-2(9bH)-one	84.8
6	6.777	1.683	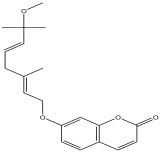	7-((2E,5E)-7-methoxy-3,7-dimethylocta-2,5-dienyloxy)-2H-chromen-2-one	84.3
7	9.888	2.19	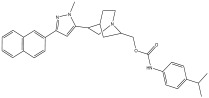	(3-(1-methyl-3-(naphthalen-6-yl)-1H-pyrazol-5-yl)quinuclidin-7-yl)methyl 4-isopropylphenylcarbamate	81.5
8	10.672	0.252	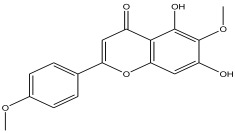	5,7-dihydroxy-6-methoxy-2-(4-methoxyphenyl)-4H-chromen-4-one	81.2
9	12.567	0.945	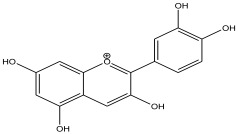	Cyanidin	89.2
10	13.293	1.414	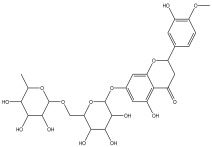	No name identity from NIST Library	95.7
11	14.781	2.77	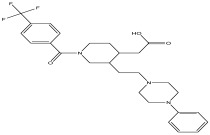	No name identity from NIST library	90.6
12	16.327	6.629	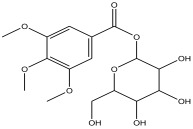	tetrahydro-3,4,5-trihydroxy-6-(hydroxymethyl)-2H-pyran-2-yl 3,4,5-trimethoxybenzoate	89.1
13	17.859	0.823	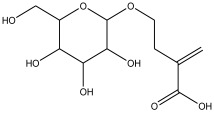	2-methylene-4-(tetrahydro-3,4,5-trihydroxy-6-(hydroxymethyl)-2H-pyran-2-yloxy) butanoic acid	80.1
14	19.133	1.078	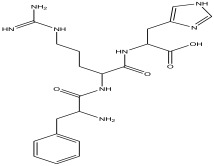	2-(guanidine)-3-(1H-imidazol-4-yl) propanoic acid	99.4
15	20.311	1.654	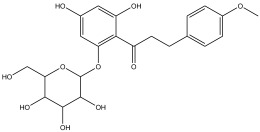	No name identity equivalent in NIST library	93.8
16	22.047	4.224	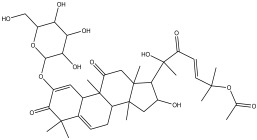	No name identity equivalent in NIST library	80.1
17	22.926	0.907	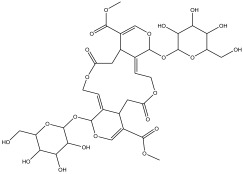	No name identity equivalent in NIST library	90.6
18	29.49	3.729	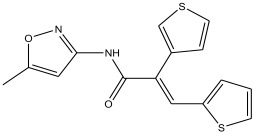	(E)-N-(5-methylisoxazol-3-yl)-3-(thiophen-2-yl)-2-(thiophen-3-yl)acrylamide	95.8
19	30.378	1.804	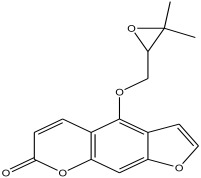	4-((3,3-dimethyloxiran-2-yl)methoxy)-7H-furo[3,2-g]chromen-7-one	88.9
20	34.241	4.11	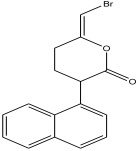	(Z)-6-(bromomethylene)-tetrahydro-3-(naphthalen-5-yl)pyran-2-one	91.2
21	42.28	2.368	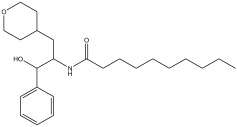	N-(3-(tetrahydro-2H-pyran-4-yl)-1-hydroxy-1-phenylpropan-2-yl)decanamide	84.6
22	47.231	11.648	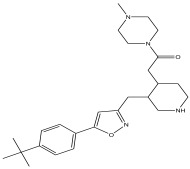	2-(3-((5-(4-tert-butylphenyl)isoxazol-3-yl) methyl) piperidin-4-yl)-1-(4-methylpiperazin-1-yl)ethanone	85.3
23	52.518	11.169	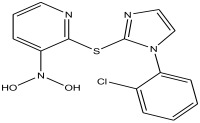	2-(1-(2-chlorophenyl)-1H-imidazol-2-ylthio)-N,N-dihydroxypyridin-3-amine	87.9
24	58.133	10.802	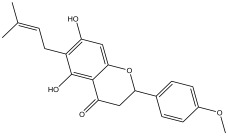	2,3-dihydro-5,7-dihydroxy-2-(4-methoxyphenyl)-6-(3-methylbut-2-enyl)chromen-4-one	85.7

* RT = retention time, ^#^ PA = peak area.

## Data Availability

All data related to this work are presented in the manuscript.

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
