# Peer review of "Green Synthesis of Characterized Silver Nanoparticle Using Cullen tomentosum and Assessment of Its Antibacterial Activity"

_antibiotics, 2023, doi:10.3390/antibiotics12020203_

Round 1

Reviewer 1 Report

The article entitled " Green synthesis of characterized silver nanoparticle using Cullen tomentosum and assessment of its antibacterial activity" is written as per the Journal style and presented nicely.

However, a few questions related to methodology requires answers. There are a few missing points in the manuscript and annotated in the PDF file (like solvent, concentration, identification of compounds, elution pattern of compound, MS spectra and similarity index). 

Reviewer 2 Report

The paper “Green synthesis of characterized silver nanoparticle using Cullen tomentosum and assessment of its antibacterial activity” by John Awungnjia Asong et al. They have green synthesized silver nanoparticle (CTAgNP) using Cullen tomentosum (Thunb.) J.W. Grimes acetone extract and then evaluated for the antibacterial activity of the plant extract and biogenic nanoparticles against two Gram-positive bacteria strains namely, Bacillus cereus and Staphylococcus aureus.

1. The introduction is very short. Go through the following reference to get additional details in this regard.

 Soliman, W.E.; Khan, S.; Rizvi, S.M.D.; Moin, A.; Elsewedy, H.S.; Abulila, A.S.; Shehata, T.M. Therapeutic Applications of Biostable Silver Nanoparticles Synthesized Using Peel Extract of Benincasa hispida: Antibacterial and Anticancer Activities. Nanomaterials 202010, 1954. https://doi.org/10.3390/nano10101954

2. Provide a UV and FTIR spectrum alone of plant extract

3. The size of nanoparticles mentioned in the abstract and the results are different justification for this discrepancy needs to addressed.

4. A valid description for the higher difference in the particle size estimated by TEM & DLS should added. Following reference may be cited to justify the obtained result.

Al Hagbani, T.; Rizvi, S.M.D.; Hussain, T.; Mehmood, K.; Rafi, Z.; Moin, A.; Abu Lila, A.S.; Alshammari, F.; Khafagy, E.-S.; Rahamathulla, M.; Abdallah, M.H. Cefotaxime Mediated Synthesis of Gold Nanoparticles: Characterization and Antibacterial Activity. Polymers 202214, 771. https://doi.org/10.3390/polym14040771

5. Give the mechanistic aspects for enhanced antibacterial activity of NP compared to the native plant extract

6. Write the significance of the UV peaks of 296 and 632

The manuscript is well written and is very relevant. However, it needs further refining for consideration for publication. Regards to this the above suggestion may seriously considered before acceptance
